# The Effects of 2′-Hydroxy-3,6′-Dimethoxychalcone on Melanogenesis and Inflammation

**DOI:** 10.3390/ijms241210393

**Published:** 2023-06-20

**Authors:** Sungmin Bae, Chang-Gu Hyun

**Affiliations:** Jeju Inside Agency and Cosmetic Science Center, Department of Chemistry and Cosmetics, Jeju National University, Jeju 63243, Republic of Korea; 9901bae99@naver.com

**Keywords:** B16F10, 2′-hydroxy-3,6′-dimethoxychalcone, inflammation, melanogenesis, RAW 264.7

## Abstract

In this study, we demonstrated that 2′-hydroxy-3,6′-dimethoxychalcone (3,6′-DMC) alleviated α-MSH-induced melanogenesis and lipopolysaccharides (LPS)-induced inflammation in mouse B16F10 and RAW 264.7 cells. In vitro analysis results showed that the melanin content and intracellular tyrosinase activity were significantly decreased by 3,6′-DMC, without cytotoxicity, via decreases in tyrosinase and the tyrosinase-related protein 1 (TRP-1) and TRP-2 melanogenic proteins, as well as the downregulation of microphthalmia-associated transcription factor (MITF) expression through the upregulation of the phosphorylation of extracellular-signal-regulated kinase (ERK), phosphoinositide 3-kinase (PI3K)/Akt, and glycogen synthase kinase-3β (GSK-3β)/catenin, and downregulation of the phosphorylation of p38, c-Jun N-terminal kinase (JNK), and protein kinase A (PKA). Furthermore, we investigated the effect of 3,6′-DMC on macrophage RAW264.7 cells with LPS stimulation. 3,6′-DMC significantly inhibited LPS-stimulated nitric oxide production. 3,6′-DMC also suppressed the expression of inducible nitric oxide synthase (iNOS) and cyclooxygenase (COX)-2 on the protein level. In addition, 3,6′-DMC decreased the production of the tumor necrosis factor-α and interleukin-6. Successively, our mechanistic studies revealed that 3,6′-DMC also suppressed the LPS-induced phosphorylation of the inhibitor of IκBα, p38MAPK, ERK, and JNK. The Western blot assay results showed that 3,6′-DMC suppresses LPS-induced p65 translocation from cytosol to the nucleus. Finally, the topical applicability of 3,6′-DMC was tested through primary skin irritation, and it was found that 3,6′-DMC, at 5 and 10 μM concentrations, did not cause any adverse effects. Therefore, 3,6′-DMC may provide a potential candidate for preventing and treating melanogenic and inflammatory skin diseases.

## 1. Introduction

The physiological and morphological aspects of skin undergo frequent changes. Internal and external factors directly impact the occurrence of various skin problems, such as aging, inflammation, and hyperpigmentation [1]. Melanin production can be triggered by various external factors, including exposure to ultraviolet (UV) radiation and inflammatory stimuli. Among these factors, UV radiation is recognized as the primary pathogenic factor leading to melanogenesis. Pathological melanogenesis can contribute to skin inflammation and the production of pro-inflammatory cytokines, potentially hastening the process of skin aging [2]. Inhibitors of melanin synthesis are widely used in pharmaceuticals and cosmetics and are typically tyrosinase inhibitors, such as kojic acid and its derivatives. However, these agents only target the downstream melanogenic pathway and can be accompanied by serious side effects, such as mild carcinogenicity and discoloration due to the irritability pathological compensatory function of the cells [3]. Inflammation, on the other hand, is typically managed with non-steroidal anti-inflammatory drugs, which, when taken continuously, can cause clinically significant gastrointestinal kidney damage, increased risk of cardiovascular disease, and liver damage [4].

Plant polyphenols are increasingly being used as skin health agents and gaining popularity. Plant polyphenols are recognized for their ability to boost the body’s innate immune system, offering protection against a range of skin diseases [5]. The biological activity of these polyphenols depends on their physicochemical properties, enabling them to overcome epidermal barriers and interact with specific receptors. Multiple studies have provided evidence supporting the vital role of polyphenols in alleviating various skin issues and promoting the restoration of a healthy skin condition [1]. For this reason, plant-derived polyphenols have been used to develop anti-inflammatory agents and melanogenesis inhibitors in order to alleviate skin irritation and melanogenesis, and many studies have reported that polyphenolic compounds inhibit skin inflammation and melanin production [6,7,8]. Considering this principle, it is highly necessary to conduct research that specifically explores the anti-inflammatory and melanogenic inhibitory activities of new natural compounds or their derivatives.

Historically, natural products have played an important role in cosmetic ingredient discovery, particularly for their anti-wrinkle and moisturizing properties, but also in other therapeutic areas, including skin disorders such as acne, psoriasis, and atopic diseases. Natural products offer unique properties compared to conventional synthetic molecules, bringing both benefits and challenges to the cosmetic ingredient discovery process. Natural products are characterized by tremendous scaffold diversity and structural complexity. They typically have a higher molecular mass compared to synthetic compound libraries, a greater number of sp3 carbon atoms and oxygen atoms but fewer nitrogen and halogen atoms, and a greater number of H-bond acceptors and donors. These differences can be advantageous, and their use in traditional medicine can provide insights into their efficacy and safety. Overall, the natural product pool is enriched with “bioactive” compounds that cover a larger area of chemical space compared to typical synthetic small-molecule libraries [9].

Chalcones, classified as a subclass of open-chain flavonoids, represent a potentially intriguing group of compounds in this regard. The typical structural feature of chalcones (1,3-diphenyl-2-propen-1-ones) is an open, α-β unsaturated, three-carbon fragment connecting two aromatic rings. They are yellow and occur in many naturally occurring plant-derived compounds, such as those from fruits (e.g., citruses, apples), vegetables (e.g., tomatoes, shallots, bean sprouts, potatoes), and various plants and spices (e.g., licorice) [10]. The chalcone scaffold is widely recognized as a privileged structure in medicinal chemistry, and it is frequently utilized as a highly effective template for drug discovery [11]. Due to their straightforward chemistry and convenient synthesis methods, numerous chalcone derivatives have been synthesized and exhibit a wide range of activities, including antiparasitic, antioxidant, anti-inflammatory, neuroprotective, antiulcer, antifungal, and cytotoxic activities, as well as being applied in vaccines that control human leishmaniasis in different biological systems [11,12,13,14]. Certain clinical studies have revealed that chalcone-containing compounds can attain high plasma concentrations without producing significant levels of toxicity [15]. Therefore, great interest in the promising biological profiles of chalcone derivatives has arisen in both academia and industry.

In our ongoing screening program, aimed towards repurposing classical drugs and natural compounds, we discovered that several antibiotics, flavonoids, and coumarins exhibit both anti-inflammatory and melanogenic activities, making them suitable for application in cosmeceuticals and nutraceuticals [16,17,18,19]. Building upon this study, we expanded our investigation to screen 2’-hydroxychalcone derivatives. In this study, we determined the effects of 2′-hydroxy-3,6′-dimethoxychalcone (3,6′-DMC) on inflammation and melanogenesis in mouse macrophage RAW 264.7 and B16F10 melanoma cells (Figure 1). Furthermore, we analyzed the mechanism of 3,6′-DMC.

## 2. Results

### 2.1. Effects of 3,6′-DMC on the Viability, Melanin Content, and Tyrosinase Activity of B16F10 Cells

To investigate the concentration at which 3,6′-DMC does not affect cell viability in B16F10 cells, an MTT assay was performed. B16F10 cells were treated with various concentrations (2.5–40 μM) of 3,6′-DMC and incubated for 72 h. The cell viability criterion was unaffected if the cell viability was greater than 90% compared to the untreated control. The experimental results showed that 3,6′-DMC did not show cytotoxicity at concentrations below 5 μM (Figure 2a). Therefore, further experiments were conducted at concentrations below 5 μM, which did not show cytotoxicity. To investigate the effects of 3,6′-DMC on melanin production and tyrosinase activity, B16F10 melanoma cells were treated with different concentrations (1.25, 2.5, and 5 μM) of 3,6′-DMC and cultured for 72 h. Alpha-melanocyte-stimulating hormone (α-MSH, 100 nM) and arbutin (200 μM) were used as negative and positive controls, respectively. As shown in Figure 2b,c, 3,6′-DMC inhibited melanin production and tyrosinase activity in the α-MSH-induced B16F10 cells in a concentration-dependent manner, with a concentration of 5 μM showing almost as much inhibitory activity as the positive control, arbutin, at 200 μM. Compared to the negative control (α-MSH), the results showed that 3,6′-DMC inhibited melanogenesis by approximately 31.42% at 5 μM compared to α-MSH alone, and 3,6′-DMC inhibited tyrosinase activity by approximately 37.97% compared to α-MSH alone.

### 2.2. 3,6′-DMC Regulated the Expression of Melanogenesis-Related Proteins in B16F10 Cells

Enzymes that promote melanin synthesis, such as tyrosinase, tyrosinase-related protein 1 (TRP-1), and TRP-2, play important roles in the melanogenesis process, and MITF is a transcription factor for melanogenic enzymes, which regulates the growth, differentiation, and function of melanocytes by activating enzymes involved in melanogenesis. Therefore, Western blots were performed to investigate whether 3,6′-DMC affects the expression of these melanogenesis-related proteins in α-MSH-stimulated B16F10 cells. As shown in Figure 3, the protein expression of tyrosinase, TRP-1, and TRP-2 induced by α-MSH was significantly reduced by 3,6′-DMC in a concentration-dependent manner. Compared with the α-MSH-alone treatment group, tyrosinase, TRP-1, and TRP-2 were reduced by approximately 36.26%, 26.36%, and 46.50% at a 5 μM concentration, respectively. To investigate the influence of microphthalmia-associated transcription factor (MITF) on these melanogenic enzyme inhibitory effects of 3,6′-DMC, we checked MITF protein expression. As shown in Figure 4, the expression of MITF induced by α-MSH was significantly inhibited in a concentration-dependent manner, and 3,6′-DMC reduced it by approximately 46.62% at 5 μM compared with the α-MSH-alone treatment group. Therefore, the melanogenic enzyme expression of 3,6′-DMC is inhibited by downregulation of the expression level of MITF, a transcriptional regulator.

### 2.3. 3,6′-DMC Inhibited Melanogenesis in B16F10 Cells through the Protein Kinase A (PKA)/cAMP-Response-Element-Binding Protein (CREB) Signaling Pathway

Within the cAMP/PKA pathway, the binding of α-MSH to melanocortin-1 receptor (MC1R) directly triggers the activation of MC1R. This activation leads to the accumulation of intracellular cyclic adenosine monophosphate (cAMP), which subsequently induces the phosphorylation of PKA via cAMP. The phosphorylated PKA then translocates into the nucleus, initiating a cascade of events that involve the phosphorylation and activation of CREB. The activated CREB ultimately upregulates the transcription of MITF, a key regulator known for its role in promoting melanogenesis [20,21]. Therefore, we next performed Western blots to investigate whether 3,6′-DMC inhibits melanogenesis through the PKA/CREB signaling pathway in α-MSH-stimulated B16F10 cells. The results showed that the α-MSH-induced phosphorylation of CREB and PKA was significantly reduced by 3,6′-DMC in a concentration-dependent manner. Specifically, compared to α-MSH (100 nM) alone, the phosphorylated CREB and PKA were reduced by approximately 24.22% and 24.77% at 5 μM, respectively (Figure 5). Thus, 3,6′-DMC downregulates the expression level of MITF, a transcriptional regulator, through the cAMP/PKA signaling pathway, which ultimately inhibits melanogenesis.

### 2.4. 3,6′-DMC Repressed Melanogenesis in B16F10 Cells through Phosphoinositide 3-Kinase (PI3K)/AKT Signaling Pathways

The close association between the PI3K/AKT pathway and glycogen synthase kinase-3β (GSK3β) has been well-established in previous studies, highlighting the role of activated AKT in phosphorylating GSK3β at Ser 9. This phosphorylation event leads to the inactivation of GSK3β, subsequently preventing the degradation of β-catenin [22,23]. To explore the potential inhibitory effect of 3,6′-DMC on melanogenesis, Western blot analyses were conducted on α-MSH-stimulated B16F10 cells to investigate whether 3,6′-DMC impacts the PI3K/AKT signaling pathway. The results showed that the phosphorylation of AKT induced by α-MSH was significantly reduced by 3,6′-DMC in a concentration-dependent manner. Compared with the α-MSH (100 nM)-alone treatment group, phosphorylated AKT was reduced by approximately 65.45% at 5 μM (Figure 6). Thus, 3,6′-DMC downregulates the expression of MITF, a transcriptional regulator, through the PI3K/AKT signaling pathway, which ultimately inhibits melanogenesis.

### 2.5. 3,6′-DMC Suppressed Melanogenesis in B16F10 Cells through the Mitogen-Activated Protein Kinase (MAPK) Signaling Pathway

In the MAPK pathway, the phosphorylation of extracellular-signal-regulated kinase (ERK) serves as a negative signaling event which, in turn, leads to the phosphorylation of MITF, resulting in the downregulation of melanogenesis. Conversely, the phosphorylation of p38 and c-Jun N-terminal kinase (JNK) enhances the expression of MITF, thus promoting melanogenesis [22,23]. With this in mind, Western blot analyses were conducted to examine the potential inhibitory effects of 3,6′-DMC on melanogenesis through the MAPK signaling pathway in α-MSH-stimulated B16F10 cells. The results showed that the phosphorylation of ERK, a negative signaling pathway, was significantly increased with increasing concentrations of 3,6′-DMC, while the phosphorylation of p38 and JNK, positive signaling pathways, was significantly decreased (Figure 7). Compared with the α-MSH (100 nM)-alone treatment group, phosphorylated ERK at 5 μM increased by approximately 78.15%, and phosphorylated p38 and JNK decreased by approximately 28.17% and 53.10%, respectively. Therefore, it can be concluded that 3,6′-DMC inhibits melanogenesis by downregulating the expression level of MITF, a transcriptional regulator, through the MAPK signaling pathway.

### 2.6. 3,6′-DMC Repressed melanogenesis in B16F10 Cells through GSK-3β/β-Catenin Signaling Pathway

In the Wnt/β-catenin pathway, the activation of a membrane receptor protein occurs upon binding with Wnt ligands. This activation leads to the phosphorylation of GSK3β at Ser 9, resulting in the inactivation of GSK3β. Consequently, β-catenin is released from the cytoplasm and translocates into the nucleus, ultimately leading to an increase in the expression of MI [24,25]. To investigate the potential impact of 3,6′-DMC on melanogenesis, Western blot analysis was performed on α-MSH-stimulated B16F10 cells, focusing on the Wnt/β-catenin signaling pathway. Therefore, Western blot was performed to investigate whether 3,6′-DMC inhibits melanogenesis through the Wnt/β-catenin signaling pathway in B16F10 cells stimulated with α-MSH. The results showed that the expression of β-catenin and phosphorylation of GSK3β induced by α-MSH were significantly decreased by 3,6′-DMC in a concentration-dependent manner, and the phosphorylation of β-catenin induced by α-MSH was increased in a concentration-dependent manner. Compared with the α-MSH (100 nM)-alone treatment group, the 5 μM concentration decreased β-catenin and phosphorylated GSK3β by approximately 17.55% and 56.03%, respectively, and increased phosphorylated β-catenin by approximately 23.98% (Figure 8). Thus, it can be seen that 3,6′-DMC downregulates the expression level of MITF, a transcriptional regulator, through the Wnt/β-catenin signaling pathway, which ultimately inhibits melanogenesis.

### 2.7. Effects of 3,6′-DMC on the Viability of Nitric Oxide (NO) and Pro-Inflammatory Cytokine Production in RAW 264.7 Cells

To investigate the concentration at which 3,6′-DMC does not affect cell viability in RAW 264.7 cells, an MTT assay was performed. RAW 264.7 cells were treated with various concentrations (2.5–40 μM) of 3,6′-DMC and incubated for 24 h. The cell viability criterion was unaffected if the cell viability was greater than 90% compared to the untreated control. The results showed that 3,6′-DMC was not cytotoxic at concentrations below 10 μM. To investigate the effect of 3,6′-DMC on NO production in RAW 264.7 cells at concentrations below 10 μM, a range that does not show cytotoxicity, NO production measurement experiments were performed. RAW 264.7 cells were treated with various concentrations (1.25–10 μM) of 3,6′-DMC and incubated for 24 h. The iNOS inhibitor L-NIL (40 μM) was used as a positive control. The extent to which NO production was inhibited compared to the LPS (1 μg/mL)-alone treatment group was investigated. The results showed that at 10 μM, 3,6′-DMC inhibited the production of NO by approximately 72.58% compared to LPS alone, as shown in (Figure 9). Next, we investigated whether 3,6′-DMC inhibits PGE_2_ and inflammatory cytokines (IL-6, IL-1β, and TNF-α) in LPS-stimulated RAW 264.7 cells. Our results showed that 3,6′-DMC inhibited IL-6 and TNF-α production in a concentration-dependent manner. However, 3,6′-DMC pretreatment had no significant effect on IL-1β production compared to the control group in this assay (Figure 10).

### 2.8. 3,6′-DMC Inhibited the Expression of Inducible Nitric Oxide Synthase (iNOS) and Cyclooxygenase-2 (COX-2) Proteins in RAW 264.7 Cells

To investigate whether the inhibition of NO and PGE_2_ production in RAW 264.7 cells by 3,6′-DMC at concentrations below 10 μM, the range in which it is not cytotoxic, was due to the downregulation of iNOS and COX-2 expression, Western blots were performed. The results showed that the expression of iNOS and COX-2 induced with LPS was inhibited by 3,6′-DMC in a concentration-dependent manner (Figure 11). Compared with LPS (1 μg/mL) alone, the 10 μM concentration reduced iNOS by 83.21% and COX-2 by 16.72%, suggesting that the 3,6′-DMC-induced reduction in iNOS and COX-2 expression may lead to the decreased production of the inflammatory mediators NO and PGE_2_.

### 2.9. 3,6′-DMC Inhibited Inflammation in RAW 264.7 Cells through the MAPK Signaling Pathway

The inhibition of the activation of the MAPK signaling pathway in LPS-stimulated macrophages results in a decrease in the production of various proinflammatory cytokines and inflammatory mediators, resulting in an anti-inflammatory effect [26,27]. Therefore, Western blots were performed to investigate whether the inhibition of NO and proinflammatory cytokine production by 3,6′-DMC in LPS-stimulated RAW 264.7 cells was due to the MAPK signaling pathway. The results showed that the phosphorylation of ERK and p38 induced by LPS was inhibited by 3,6′-DMC in a concentration-dependent manner. Compared with the LPS (1 μg/mL)-alone treatment group, phosphorylated ERK and p38 were reduced by approximately 25.80% and 67.31%, respectively, at 10 μM (Figure 12). Thus, it can be seen that 3,6′-DMC inhibits inflammation by downregulating the production levels of NO and pro-inflammatory cytokines through inhibition of the activation of the MAPK signaling pathway.

### 2.10. 3,6′-DMC Repressed Inflammation in RAW 264.7 Cells through Nuclear Factor κB (NF-κB) Signaling Pathways

NF-κB exists in an inactive state in the cytoplasm through binding to inhibitor kappa B-α (IκB-α), and the activation of NF-κB occurs in LPS-stimulated macrophages after IκB-α is phosphorylated and degraded. It has been reported that activated NF-κB translocates from the cytoplasm to the nucleus and activates the members of various proinflammatory cytokines and inflammatory transcription factors [28,29]. Therefore, Western blot was performed to investigate whether the inhibition of proinflammatory cytokine and inflammatory mediator production by 3,6′-DMC in LPS-stimulated RAW 264.7 cells was due to the NF-κB signaling pathway. The results showed that the LPS-induced phosphorylation of IκB-α was inhibited by 3,6′-DMC in a concentration-dependent manner. Consistent with these results, 3,6′-DMC also inhibited the degradation of IκB-α in the cytoplasm in a concentration-dependent manner. Compared with LPS (1 μg/mL) alone, the 10 μM concentration decreased phosphorylated IκB-α by approximately 51.89% and increased NF-κB-bound IκB-α by approximately 40.6%. Thus, 3,6′-DMC inhibits inflammation by inhibiting the degradation of IκB-α through the regulation of its phosphorylation. To investigate the translocation of p65 from the cytoplasm to the nucleus in LPS-stimulated RAW 264.7 cells, Western blots were performed. The results showed that the production of p65 in the cytoplasm was increased by 3,6′-DMC in a concentration-dependent manner. Consistent with these results, 3,6′-DMC inhibited the production of p65 in the nucleus in a concentration-dependent manner. Compared with LPS (1 μg/mL) alone, the 10 μM concentration increased p65 in the cytoplasm by approximately 29.33% and decreased p65 in the nucleus by approximately 57.80%. Therefore, 3,6′-DMC inhibits inflammation by preventing the nuclear translocation of NF-κB.

### 2.11. Skin Primary Irritation Test

Patches containing 3,6′-DMC at concentrations of 5 and 10 µM were applied to the skin and left in contact for 24 h. After removing the patch, the area was observed 48 h later. The results, as presented in Table 1, categorized the test substance (3,6′-DMC) as “none to slight” in terms of its effects. Squalene was used as a negative control for comparison.

## 3. Discussion

Chalcone and its derivatives are abundant in edible and medicinal plants and have been studied for their multiple biological activities [11,12,13,14]. Although compounds with various pharmacological properties have been developed based on the chalcone skeleton, the anti-melanogenic and anti-inflammatory effects of 2′-hydroxy-3,6′-dimethoxychalcone (3,6′-DMC) have not been fully investigated. Therefore, we aimed to determine the biological activity of 3,6′-DMC in B16F10 and RAW 264.7 cells.

Melanocytes located in the epidermis deliver melanin to neighboring keratinocytes to form the pigmented skin layer. Two types of melanin exist in mammals: the first, being universal, is eumelanin, which is black-brown in color, and the second is pheomelanin, which is red-yellow in color; in both cases, they are produced from tyrosine via a tyrosinase-dependent pathway. This melanogenic process triggers the activation of the multiplayer transcription factor MITF, which regulates tyrosinase, TRP-1, and TRP-2. Therefore, the blocking of tyrosinase-related signaling pathways prevents the early stages of melanogenesis and benefits the skin-whitening response [30,31]. Previously, hydroquinone, arbutin, and kojic acid were used as inhibitors of melanogenesis for hyperpigmentation. However, hydroquinone is mutagenic and carcinogenic to cells and has been associated with various side effects, such as phototoxicity, contact dermatitis, and irritation. Arbutin, a glucose attachment to hydroquinone, acts as a tyrosinase inhibitor, reducing or inhibiting melanin synthesis. Nevertheless, arbutin is chemically unstable in its natural form and can release toxic hydroquinone when separated from hydroquinone and glucose. Kojic acid can lead to side effects such as contact dermatitis, sensitization, and erythema, and its use has been restricted due to its storage instability [32,33,34]. Consequently, there is a need to develop melanogenesis inhibitors that effectively hinder melanogenesis with fewer potential risks and side effects for humans. Consequently, experiments were carried out to determine the inhibitory activity of 3,6′-DMC towards melanogenesis, and its efficacy was demonstrated.

The first objective of this study was to investigate the molecular mechanism underlying the inhibitory effect of 3,6′-DMC on melanogenesis in B16F10 cells. Initially, we assessed the potential cytotoxicity of 3,6′-DMC to B16F10 cells and observed no significant cytotoxic effects within the range of 1.25–5 μM. Subsequently, we discovered that 3,6′-DMC exhibited a concentration-dependent suppression of both melanin synthesis and tyrosinase activity. These findings indicate that 3,6′-DMC reduces cellular melanin synthesis in B16F10 cells by downregulating tyrosinase activity. To gain insight into the molecular mechanisms underlying the inhibitory effects of 3,6′-DMC, we examined its effects on the expression of melanogenesis-related proteins. Our results showed that 3,6′-DMC treatment led to a significant and concentration-dependent decrease in the expression levels of MITF, tyrosinase, TRP-1, and TRP-2. Therefore, these results suggest that 3,6′-DMC indirectly inhibits tyrosinase activity by downregulating MITF and other melanogenesis-related proteins in B16F10 cells.

Previous studies have shown that melanogenesis, the process of producing melanin, is influenced by the cAMP signaling pathway. Specifically, genes involved in melanogenesis, such as MITF, tyrosinase, TRP-1, and TRP-2, are activated through the cAMP-PKA-CREB pathway [20,21]. Our study of 3,6′-DMC showed a significant decrease in CREB and PKA phosphorylation after 3,6′-DMC treatment, suggesting that 3,6′-DMC affects melanogenesis through the cAMP-PKA-CREB signaling pathway. However, we need additional data on the effects of 3,6′-DMC on the cAMP/PKA signaling pathway, particularly by assessing the level of direct cAMP production, so that the effects of 3,6′-DMC on melanogenesis can be studied further. Previous studies have shown that multiple signaling pathways are involved in the regulation of melanin synthesis, and in particular, signaling pathways involving MAPKs, including p38 MAPK, ERK, and JNK, and PI3K/AKT, are involved in the regulation of MITF expression [22,23]. As shown in the results in Figure 7, we demonstrate that 3,6′-DMC inhibits melanogenesis in B16F10 cells by increasing P-ERK and decreasing P-JNK and P-P38 levels, providing evidence that 3,6′-DMC activates melanogenesis through the MAPK signaling pathway. To further evaluate the effect of 3,6′-DMC on the MAPK signaling pathway, we also assessed the phosphorylation status of AKT using Western blot analysis. As expected, 3,6′-DMC activated AKT phosphorylation. These findings convincingly demonstrate that 3,6′-DMC inhibits melanogenesis by modulating p-AKT (Figure 6).

Furthermore, we explored the potential involvement of the Wnt/β-catenin signaling pathway in melanin synthesis. Our findings demonstrated that treatment with 3,6′-DMC led to an upregulation of P-β-catenin, while it downregulated p-GSK-3β and β-catenin. GSK-3β, acting as a negative regulator within the Wnt/β-catenin pathway, can activate MITF’s function by phosphorylating it at Ser 298. Additionally, activated GSK-3β induces the phosphorylation of the N-terminal Ser/Thr residues in β-catenin, resulting in the ubiquitination and degradation of β-catenin. Melanin synthesis is regulated through intricate crosstalk between various signaling pathways [24,25]. Therefore, these outcomes suggest that the inhibitory effects of 3,6′-DMC on melanogenesis are associated with the activation of the PI3K/Akt or ERK signaling pathways, along with the inactivation of the Wnt/β-catenin and PKA/CREB signaling pathways.

Excessive and persistent inflammation can cause a variety of inflammatory diseases that affect multiple organs, including atherosclerosis, allergies, asthma, chronic obstructive pulmonary disease, inflammatory bowel disease, psoriasis, and rheumatoid arthritis [35,36]. Non-steroidal anti-inflammatory drugs (NSAIDs) are commonly used to relieve pain and inflammation, but they are associated with a variety of side effects. Some patients who overdosed on NSAIDs have been reported to experience serious clinical complications, such as gastrointestinal upset, coma, acute renal failure, and cardiovascular disease [37,38]. Therefore, extensive research has been conducted to find safe and effective anti-inflammatory drugs for human use. As part of this effort, experiments were conducted to evaluate the anti-inflammatory activity of 3,6′-DMC and demonstrate its effectiveness.

To evaluate the anti-inflammatory properties of 3,6′-DMC, we first performed an MTT assay to determine the cell viability concentrations. Concentrations with cell permeability greater than 90% compared to the untreated group were selected for further analysis. We then measured NO production at non-cytotoxic concentrations. The results showed that 3,6′-DMC was not cytotoxic at concentrations below 10 μM; therefore, subsequent experiments were performed using concentrations below this threshold. LPS-stimulated macrophages are known to produce the pro-inflammatory enzyme iNOS, which acts as a catalyst for the conversion of L-arginine to NO, thereby promoting various inflammatory processes associated with conditions such as septic shock and rheumatoid arthritis. In conclusion, we investigated the effect of 3,6′-DMC on NO production and found that at a concentration of 10 μM, 3,6′-DMC reduced NO production by 72.58% compared to the LPS-alone treatment group. Notably, this inhibition of NO production surpassed the positive control, L-NIL. To explore the potential of 3,6′-DMC as an inhibitor of the inflammatory response, we used an ELISA kit to measure the production of PGE_2_, a key mediator of inflammation. The results showed that PGE_2_ production induced by LPS stimulation was slightly inhibited in the presence of 3,6′-DMC. We also investigated the effect of 3,6′-DMC on the production of the pro-inflammatory cytokines IL-6, IL-1β, and TNF-α, which are typically induced by LPS. Specifically, as shown in Figure 10, 3,6′-DMC inhibited the production of IL-6 and TNF-α but had a negligible effect on the production of PGE_2_ and IL-1β.

Macrophages activated via LPS stimulation activate the MAPK signaling pathway, including ERK, p38, and JNK, to produce a variety of proinflammatory cytokines and inflammatory mediators. Therefore, we next performed Western blot experiments to elucidate the involvement of MAPK signaling as a mechanism through which 3,6′-DMC inhibits the inflammatory response of RAW 264.7 cells [30,31]. As shown in Figure 12, the phosphorylation of ERK, JNK, and p38 by LPS in RAW 264.7 cells was attenuated in the presence of 3,6′-DMC, indicating the ability of 3,6′-DMC to downregulate the production of pro-inflammatory cytokines and inflammatory mediators through the MAPK signaling pathway. In macrophages, NF-κB remains inactive in the cytoplasm when bound to IκB-α, but upon exposure to external stimuli such as LPS, it is degraded and released with the phosphorylation of IκB-α, which activates NF-κB. Subsequently, the activated NF-κB translocates from the cytoplasm to the nucleus and triggers the activation of various pro-inflammatory cytokines and pro-inflammatory transcription factors [32,33]. However, here, this phenomenon was reversed after 3,6′-DMC treatment, through which we observed that the phosphorylation of IκB-α was reduced and the degradation of IκB-α induced by LPS was inhibited in RAW 264.7 cells, resulting in the increased binding of NF-κB to IκB-α (Figure 13). We also examined the translocation of p65, a subunit of NF-κB, from the cytoplasm to the nucleus and found that 3,6′-DMC promoted the production of p65 in the cytoplasm while inhibiting its production in the nucleus. These findings suggest that 3,6′-DMC prevents the translocation of p65 from the cytoplasm to the nucleus, thereby interfering with the phosphorylation and degradation of IκB-α, ultimately inhibiting inflammation (Figure 14). Finally, 3,6′-DMC was evaluated for potential application as a topical ingredient using a human skin primary irritation test. Both 5 μM and 10 μM concentrations of 3,6′-DMC were tested on the backs of 33 subjects. The results showed that the test material was in the low-irritation category in terms of human skin primary irritation. This suggests that 3,6′-DMC is a safe ingredient for topical application.

In conclusion, 3,6′-DMC, at a low concentration of 5 μM, reduced the melanin content and inhibited the intracellular tyrosinase activity of α-MSH-stimulated B16F10 cells to a degree comparable to the effect of the positive control, an arbutin concentration of 200 μM. It also reduced the expression of melanogenic enzymes such as tyrosinase, TRP-1, and TRP-2, and ultimately inhibited melanogenesis by reducing the expression of MITF through PKA/CREB, PI3K/AKT, MAPK, and Wnt/β-catenin signaling pathways. Furthermore, we investigated the anti-inflammatory activity of 3,6′-DMC in LPS-stimulated RAW 264.7 cells and found that treatment with 3,6′-DMC dramatically inhibited NO production, decreased the expression of the inflammatory proteins iNOS and COX-2, and suppressed the production of pro-inflammatory mediators and pro-inflammatory cytokines through the MAPK and NF-κB signaling pathways. Based on these results, 3,6′-DMC can be considered a promising strategy for the development of topical therapeutics and whitening agents for the treatment of hyperpigmentation, improving anti-inflammatory therapy and inflammatory diseases.

## 4. Materials and Methods

### 4.1. Chemicals and Reagents

The Dulbecco’s modified Eagle’s medium (DMEM), penicillin/streptomycin, 10Ⅹ trypsin-EDTA (0.5%), and bicinchoninic acid (BCA) protein kits used in this study were purchased from Thermo Fisher Scientific (Waltham, MA, USA). Fetal bovine serum (FBS) was purchased from Merck Millipore (Burlington, MA, USA), and 3-(4,5-dimethylthiazol-2-yl)-2,5-diphenyltetrazolium bromide (MTT), dimethyl sulfoxide (DMSO), and sodium dodecyl sulfate (SDS) were used. Triple-buffered saline (TBS), phosphate-buffered saline (PBS), radioimmunoprecipitation assay (RIPA) buffer, and an enhanced chemiluminescence (ECL) kit were purchased from Biosesang (Seongnam, Gyeonggi, Korea). 3,6′-Dimethoxy-2-hydroxychalcone, *Escherichia coli*-derived lipopolysaccharide (LPS), α-melanocyte-stimulating hormone (α-MSH), grease reagent, a protease inhibitor cocktail, sodium hydroxide (NaOH), L-3,4-dihydroxyphenylalanine (L-DOPA), sodium phosphate monobasic, and sodium phosphate dibasic were purchased from Sigma-Aldrich (St. Louis, MO, USA). L-N6-(1-iminoethyl)lysine dihydrochloride (L-NIL) and N-[2-(cyclohexyloxy)-4-nitrophenyl]methanesulfonamide (NS-398) were purchased from Cayman Chemical Company (Ann Arbor, MI, USA), while 2× Laemmli sample buffer and Tween 20 were purchased from Bio-Rad (Hercules, CA, USA), skimmed milk powder from BD Difco (Sparks, MD, USA), and bovine serum albumin (BSA) was obtained from Bovostar (Melbourne, Australia). Of the ELISA kits, PGE_2_ was purchased from Abcam (Cambridge, UK), and TNF-α, IL-1β, and IL-6 were obtained from BD Biosciences (Franklin Lakes, NJ, USA). Among the primary antibodies used for Western blots, tyrosinase, TRP-1, TRP-2, MITF, p-CREB, CREB, and β-actin were purchased from Santa Cruz Biotechnology (Dallas, TX, USA), while the protease/phosphatase inhibitor cocktail and p-ERK, ERK, p-JNK, JNK, p-p38, p38, p-IκB-α, IκB-α, p-AKT, AKT, p-GSK-3β, GSK-3β, p-β-catenin, β-catenin, p-PKA, PKA, β-actin, p65, Lamin B, as well as secondary anti-mouse and anti-rabbit antibodies, were purchased from Cell Signaling Technology (Danvers, MA, USA). In addition, an anti-iNOS antibody was purchased from Merck Millipore (Burlington, MA, USA), and an anti-COX-2 antibody was purchased from BD Biosciences (Franklin Lakes, CA, USA).

### 4.2. Cell Cultures

RAW 264.7 murine macrophage cells were obtained from the Korean Cell Line Bank (KCLB), and B16F10 mouse melanoma cells were obtained from The Global Bioresource Center (ATCC). Cell cultures were grown in a DMEM medium supplemented with 10% FBS and 1% penicillin–streptomycin (P/S) at 37 °C and 5% CO_2_ for 2 and 3 days, respectively.

### 4.3. Cell Viability

To determine the range of cytotoxicity of the sample, cell viability was measured using the MTT reduction method. The MTT method is based on the reduction of the water-soluble, yellow-colored MTT tetrazolium to the non-water-soluble, purple-colored MTT formazan with a dehydrogenase present in the mitochondria of the cell. The amount of formazan reduced and formed is directly proportional to the number of surviving cells; therefore, a higher concentration of formazan indicates higher cell viability.

RAW 264.7 cells were seeded at 1.5 × 10^5^ cells/well in 24-well plates and pre-incubated in a CO_2_ incubator for 24 h. Then, they were treated and allowed to react for 24 h. B16F10 cells were seeded in 24-well plates at 8.0 × 10^3^ cells/well and preincubated in a CO_2_ incubator for 24 h; then, the samples were treated and allowed to react for 72 h. They were treated with 0.2 mg/mL of MTT reagent and reacted for 3 h. The medium was removed, the purple formazan crystals formed via reduction with 500 μL (B16F10 cells) or 800 μL (RAW 264.7 cells) of DMSO per well were dissolved, and the absorbance was measured at 570 nm using a spectrophotometric microplate reader.

### 4.4. Measuring Nitric Oxide (NO) Production

To determine if the sample inhibited NO production in the RAW 264.7 macrophages induced by LPS, the amount of NO in the cell culture was measured in the form of nitrite (NO_2_^−^). RAW 264.7 cells were seeded at 1.5 × 10^5^ cells/well in 24-well plates and incubated for 24 h prior to treatment under the same conditions as those outlined above for the cell culture method. The cells were treated with various concentrations of 1 μg/mL LPS and 40 μM L-NIL and allowed to react for 24 h. Then, in a 96-well plate, the treated cell culture and Griess reagent (1% sulfanilamide, 0.1% N-(1-naphthyl)ethylenediamine, 2.5% phosphoric acid) were mixed in a 1:1 ratio and reacted for 10 min, and the absorbance was measured at 540 nm using a microplate reader.

### 4.5. Measurement of Prostaglandin E (PGE)_2_ and Pro-Inflammatory Cytokine Production

To determine if the samples inhibited the production of inflammatory response mediators in RAW 264.7 macrophages induced by LPS, they were measured using an enzyme-linked immunosorbent assay (ELISA) method. RAW 264.7 cells were seeded at 1.5 × 10^5^ cells/well in 24-well plates and cultured for 24 h prior to treatment under the same conditions as those specified for the cell culture method. Various concentrations of samples were treated with 1 μg/mL LPS and allowed to react for 24 h. Then, the cell culture of each well was centrifuged at 15,000 rpm for 20 min, and the supernatant was obtained. The supernatant was used to determine the contents of PGE_2_, IL-6, IL-1β, and TNF-α according to the ELISA kit manufacturer’s assay.

### 4.6. Measuring Melanin Contents

To determine the effect of the sample on the melanin content of the B16F10 cells, a melanin content assay was used. B16F10 cells were seeded at 5 × 10^4^ cells/dish in 60 mm cell culture dishes and cultured for 24 h prior to treatment under the same conditions as described for the cell culture method. Cells were treated with various concentrations of sample with 100 nM α-MSH and allowed to react for 72 h. No treatment was used as a control; cells treated with 100 nM of α-MSH alone served as a negative control, and cells treated with 200 μM of arbutin served as a positive control. After incubation, the medium was removed, and the cells were washed twice with 1×PBS buffer and lysed using lysis buffer (RIPA buffer, 1% protease inhibitor cocktail) for 20 min at 4 °C. Then, they were transferred to 1.5 mL e-tubes and centrifuged at −8 °C and 15,000 rpm for 20 min. The supernatant was removed to obtain a pellet, and the isolated pellet was dissolved in 250 μL of 1N NaOH supplemented with 10% DMSO at 80 °C. They were transferred to 96-well plates at 50 μL each, and the absorbance was measured at 405 nm using a microplate reader. The results were calculated and expressed as a percentage relative to the α-MSH-alone treatment group.

### 4.7. Measuring Intracellular Tyrosinase Activity

B16F10 cells were seeded at 5 × 10^4^ cells/dish in 60 mm cell culture dishes and cultured for 24 h prior to treatment under the same conditions as described for the cell culture method. The cells were treated with various concentrations of sample with 100 nM α-MSH and allowed to react for 72 h. No treatment was used as a control; cells treated with 100 nM of α-MSH alone served as a negative control, and cells treated with 200 μM of arbutin served as a positive control. After incubation, the medium was removed, and the cells were washed twice with 1×PBS buffer and lysed using lysis buffer (RIPA buffer, 1% protease inhibitor cocktail) for 20 min at 4 °C. Then, they were transferred to 1.5 mL e-tubes and centrifuged at −8 °C and 15,000 rpm for 30 min to obtain supernatants. After building a standardization curve of bovine serum albumin (BSA) using the BCA protein assay kit, the protein content of the isolated supernatant was measured, and the protein was diluted in equal amounts in a 96-well plate, mixing 20 μL of protein sample and 80 μL of 2 mg/mL L-DOPA in each well. After incubation at 37 °C for 2 h, the resulting DOPA chrome was measured for absorbance at 490 nm using a microplate reader. The results were calculated and are expressed as percentages relative to the α-MSH-alone treatment group.

### 4.8. Western Blot

RAW 264.7 cells were seeded at 6.0 × 10^5^ cells/dish in 60 mm cell culture dishes and cultured for 24 h prior to treatment under the same conditions as described for the cell culture methods. After treatment with various concentrations of sample and 1 μg/mL LPS, the cells were incubated according to the expression time of the target protein. B16F10 cells were seeded at 7.0 × 10^4^ cells/dish in 60 mm cell culture dishes and cultured for 24 h prior to treatment under the same conditions as specified for the cell culture method, treated with various concentrations of sample and 100 nM α-MSH, and then cultured according to the expression time of each protein.

After incubation, the medium was removed and washed once with a cold 1×PBS buffer, and the cells were lysed using lysis buffer (RIPA buffer, 1% protease inhibitor cocktail) for 20 min at 4 °C. Then, they were transferred to 1.5 mL e-tubes and centrifuged at −8 °C and 15,000 rpm for 30 min to obtain supernatants. After building a standardization curve of bovine serum albumin (BSA) using the BCA protein assay kit, the concentration of protein in the separated supernatant was quantified, and samples for Western blot experiments were prepared by mixing the protein with 2×Laemmli sample buffer in a 1:1 ratio and heating at 100 °C for 5 min. After cooling the heated proteins, samples containing equal amounts of protein (16 μL) were loaded onto sodium dodecyl sulfate (SDS)-polyacrylamide gel at 100 V. The proteins were separated according to size using electrophoresis. The separated proteins were transferred to a polyvinylidene difluoride (PVDF) membrane and blocked with 1×TBS-T (tris-buffered saline containing 1% tween 20) containing 5% skim milk for 1 h. The membrane was washed six times with 1×TBS-T at 10 min intervals and incubated overnight at 4 °C with primary antibodies diluted at a 1:2000 ratio in 1×TBS-T. The membrane was then blocked with 1×TBS-T for 1 h. Next, the membrane was washed six times at 10 min intervals with 1×TBS-T to probe the primary antibody and reacted with the secondary antibody diluted 1:2000 in 1×TBS-T for 2 h at room temperature. After washing with 1×TBS-T 6 times at 5 min intervals, the membrane was reacted with an ECL kit to express specific proteins, and each target protein band was detected using Chemidoc (Vilber Lourmat, Collégien, France).

### 4.9. Human Skin Irritation Test

The ethical and scientific validity of this study was reviewed by the Institutional Review Board of Dermapro based on the Declaration of Helsinki, and it was conducted in accordance with ethical principles, with the voluntary consent of the subjects (IRB no. 1-220777-A-N-01-DICN23044). Thirty-three female subjects who met the exclusion and inclusion criteria participated in the entire study. The mean age of the subjects was 45.82 ± 7.84 years, with a maximum age of 53 years and a minimum age of 25 years. The purpose and method of the study, as well as adverse events, were explained to the selected subjects, and those who showed willingness to participate filled out an informed consent form and participated in the study. The test site (back) was washed with 70% ethanol, and 20 μL of the test substance was applied to the test site for 24 h. The first evaluation was performed 20 min after the removal of the application, and the second evaluation was performed 24 h later. The skin reaction results for each test substance were calculated according to the formula below. The average reactivity of each test substance calculated was determined in this way. However, according to the PCPC Guidelines, a skin reaction of +5 grade is more likely to be an allergic reaction rather than an irritant reaction; thus, the maximum grade was evaluated as +4 grade.
$$\mathrm{Response}=\frac{\sum \left(Grade\times No.of\;Responders\right)}{4\;(Maximum\;Grade)\times n\;(Total\;Subjects)}\times 100\times 1/2$$

### 4.10. Statistical Analyses

All experiment results were expressed as the mean ± standard deviation (SD) of at least three independent experiments. Statistical analyses were performed using Student’s *t*-tests or one-way ANOVA using IBM SPSS (v. 20, SPSS Inc., Armonk, NY, USA). * *p* < 0.1, ** *p* < 0.01, *** *p* < 0.001.

## 5. Conclusions

Natural chalcones exhibit a diverse array of pharmacological activities, including but not limited to anticancer, anti-inflammatory, and antibacterial effects. It is worth noting that the literature on chalcones and their derivatives with anti-melanogenic and anti-inflammatory activities is limited, with only a small number of previous reports available. In this study, we demonstrated that 3,6′-DMC alleviated α-MSH-induced melanogenesis and LPS-induced inflammation in mouse B16F10 melanoma and macrophage RAW 264.7 cells. The results showed that 3,6′-DMC did not cause any cytotoxicity and resulted in significant reductions in intracellular tyrosinase activity and melanin content in B16F10 cells. Additionally, 3,6′-DMC inhibits the expression of melanogenic enzymes such as tyrosinase, TRP-1, and TRP-2, suppressing melanin synthesis through a cAMP/PKA-dependent downregulation of MITF, a master transcription factor in melanogenesis. Furthermore, anti-melanogenic effects were exerted by 3,6′-DMC through the downregulation of the p38 and JNK signaling pathways and upregulation of the ERK and PI3K/Akt/GSK-3β cascades. In addition, the β-catenin content in the cell cytoplasm and nucleus was increased by 3,6′-DMC through a reduction in phosphorylated β-catenin. Based on the results, it can be concluded that 3,6′-DMC regulates melanogenesis through various signaling pathways, such as PKA, MAPKs, PI3K/Akt/GSK-3β, and β-catenin (Figure 15). Furthermore, after collecting all of the information regarding the anti-inflammatory effects of 3,6′-DMC identified in this study, a map of the relevant molecular pathways was constructed (Figure 16). First, 3,6′-DMC inhibited iNOS and COX-2 expression, indicating that the pro-inflammatory response is directly related to the production of NO in LPS-induced RAW 264.7 cells. Secondly, 3,6′-DMC inhibited interleukin IL-6 and TNF-α production in a concentration-dependent manner. Finally, 3,6′-DMC exhibited anti-inflammatory activity that depended on its ability to regulate the production of NO and other pro-inflammatory cytokines in LPS-induced RAW 264.7 cells through the suppression of NF-κB activation and MAPK phosphorylation. Finally, we tested the potential of 3,6′-DMC for topical applications by conducting primary human skin irritation tests. 3,6′-DMC did not induce any adverse reactions during these tests. These findings suggest that 3,6′-DMC has the potential to be a candidate for preventing and treating melanogenic and inflammatory skin diseases. Although the stability of 3,6′-DMC was confirmed, formulation studies with nano vehicles such as exosomes are needed to address issues such as the solubility of 3,6′-DMC and its vulnerability to chemical or metabolic degradation.

## Figures and Tables

**Figure 1 ijms-24-10393-f001:**
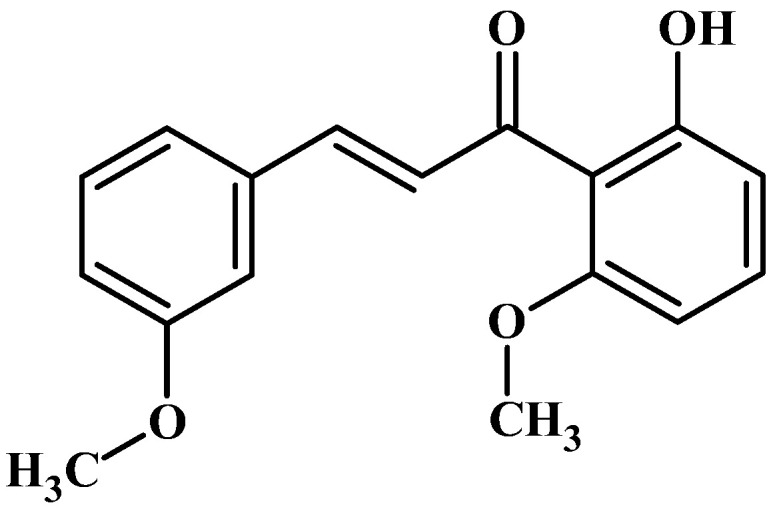
The chemical structure of 2′-hydroxy-3,6′-dimethoxychalcone (3,6′-DMC).

**Figure 2 ijms-24-10393-f002:**
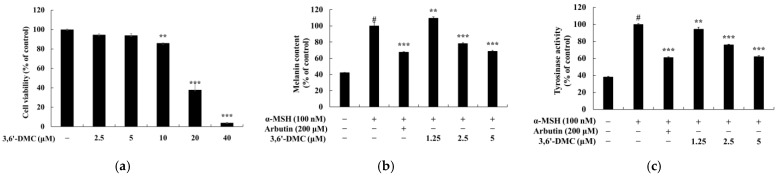
Effects of 3,6′-DMC on the cell viability (**a**), melanin content (**b**), and tyrosinase activity (**c**) of B16F10 cells. B16F10 cells were treated with different concentrations of 3,6′-DMC (2.5, 5, 10, 20, and 40 μM) for 72 h to assess cell viability. The results, expressed as the mean ± SD from three independent experiments, showed a significant impact of 3,6′-DMC on cell viability (** *p* < 0.01, *** *p* < 0.001) when compared to the untreated control group. To evaluate the melanin production and tyrosinase activity, B16F10 cells were treated with 3,6′-DMC in the presence of α-MSH (100 nM) stimulation for 72 h. Arbutin (200 μM) was utilized as the positive control. Data are expressed as the mean ± SD from three independent experiments. # *p* < 0.001 vs. untreated control group. ** *p* < 0.01, *** *p* < 0.001 vs. α-MSH-alone group.

**Figure 3 ijms-24-10393-f003:**
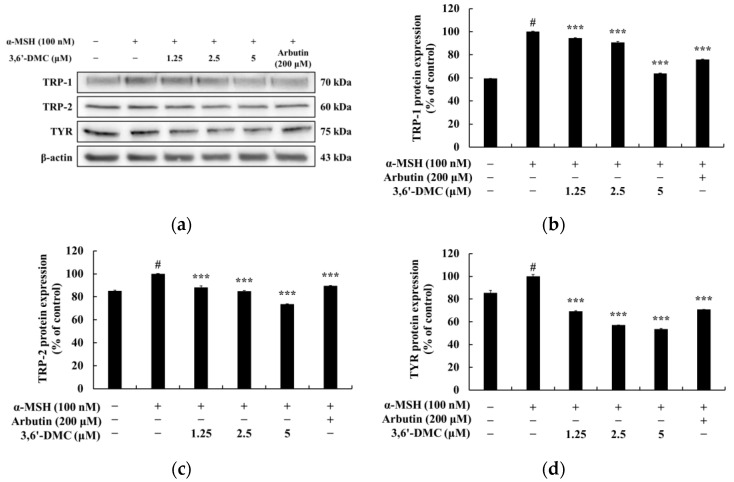
Effects of 3,6′-DMC on protein expression of TRP-1, TRP-2, and tyrosinase in α-MSH-induced B16F10 melanoma cells. The cells were treated with 3,6′-DMC (1.25, 2.5, and 5 μM) in the presence of α-MSH (100 nM) via stimulation for 24 h. (**a**) Western blotting results and protein expression of (**b**) TRP-1/β-actin, (**c**) TRP-2/β-actin, (**d**) tyrosinase/β-actin. Equal amounts of protein loading were confirmed using β-actin. Data are presented as the mean ± SD from a single triplicate experiment using ImageJ. # *p* < 0.001 vs. untreated control group. *** *p* < 0.001 vs. α-MSH-alone group.

**Figure 4 ijms-24-10393-f004:**
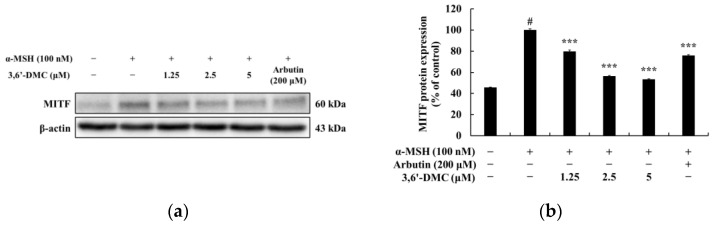
Effects of 3,6′-DMC on the protein expression of MITF in α-MSH-induced B16F10 melanoma cells. The cells were treated with 3,6′-DMC (1.25, 2.5, and 5 μM) in the presence of α-MSH (100 nM) via stimulation for 24 h. (**a**) Western blotting results and protein expression of (**b**) MITF/β-actin. Equal amounts of protein loading were confirmed using β-actin. Data are presented as the mean ± SD from a single triplicate experiment using ImageJ. # *p* < 0.001 vs. untreated control group. *** *p* < 0.001 vs. α-MSH-alone group.

**Figure 5 ijms-24-10393-f005:**
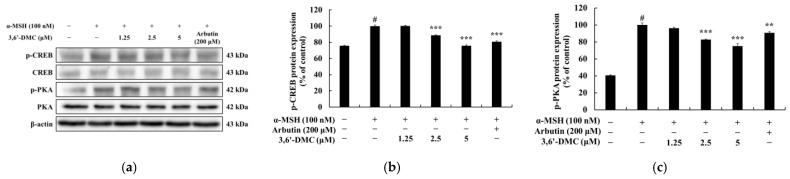
Effects of 3,6′-DMC on the phosphorylation expression of CREB and PKA in α-MSH-induced B16F10 melanoma cells. The cells were treated with 3,6′-DMC (1.25, 2.5, and 5 μM) in the presence of α-MSH (100 nM) via stimulation for 24 h. (**a**) Western blotting results and protein expression of (**b**) p-CREB/CREB and (**c**) p-PKA/PKA. Equal amounts of protein loading were confirmed using β-actin. Data are presented as the mean ± SD from a single triplicate experiment using ImageJ. # *p* < 0.001 vs. untreated control group. ** *p* < 0.01, *** *p* < 0.001 vs. α-MSH-alone group.

**Figure 6 ijms-24-10393-f006:**
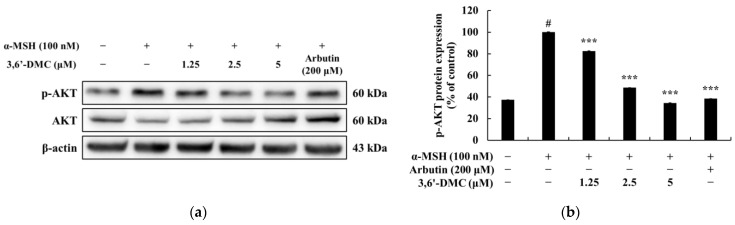
Effect of 3,6′-DMC on the phosphorylation expression of AKT in α-MSH-induced B16F10 melanoma cells. The cells were treated with 3,6′-DMC (1.25, 2.5, and 5 μM) in the presence of α-MSH (100 nM) via stimulation for 4 h. Arbutin (200 μM) was used as the positive control. (**a**) Western blotting results and protein expression of (**b**) p-AKT/AKT. Equal amounts of protein loading were confirmed using β-actin. Data are expressed as the mean ± SD from a single triplicate experiment using ImageJ software. # *p* < 0.001 vs. untreated control group. *** *p* < 0.001 vs. α-MSH-alone group.

**Figure 7 ijms-24-10393-f007:**
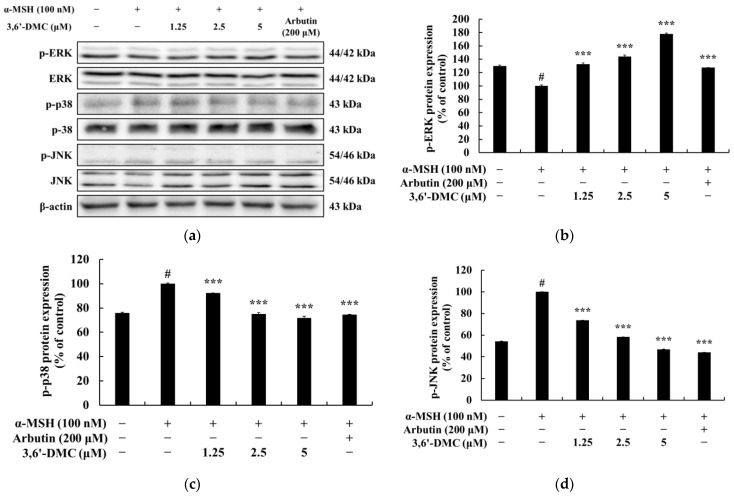
Effect of 3,6′-DMC on the phosphorylation of MAPK in α-MSH-induced B16F10 melanoma cells. The cells were treated with 3,6′-DMC (1.25, 2.5 and 5 μM) in the presence of α-MSH (100 nM) via stimulation for 4 h. Arbutin (200 μM) was used as the positive control. (**a**) Western blotting results and protein expression of (**b**) p-EKR/EKR, (**c**) p-p38/p38, (**d**) p-JNK/JNK. Equal amounts of protein loading were confirmed using β-actin. Data are expressed as the mean ± SD from a single triplicate experiment using ImageJ software. # *p* < 0.001 vs. untreated control group. *** *p* < 0.001 vs. α-MSH-alone group.

**Figure 8 ijms-24-10393-f008:**
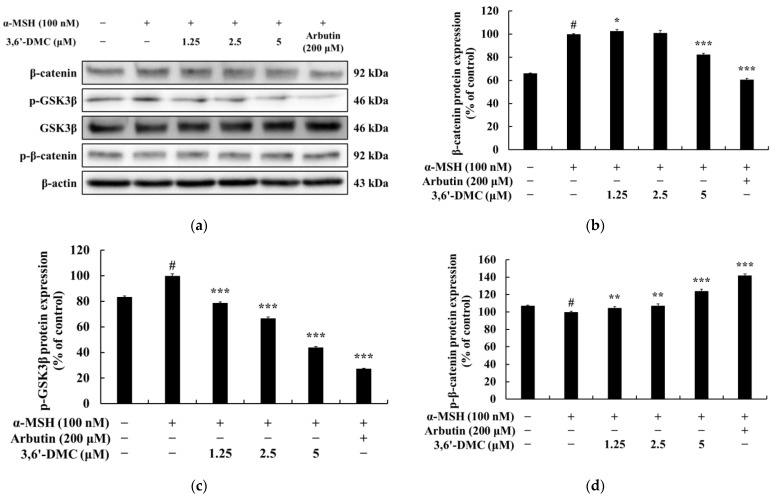
Effect of 3,6′-DMC on the protein expression of β-catenin and p-GSK3β in α-MSH-induced B16F10 melanoma cells. The cells were treated with 3,6′-DMC (1.25, 2.5, and 5 μM) in the presence of α-MSH (100 nM) via stimulation for 24 h. Arbutin (200 μM) was used as the positive control. (**a**) Western blotting results and protein expression of (**b**) β-catenin/β-actin, (**c**) p-GSK3β/GSK3β, (**d**) p-β-catenin/β-actin. Equal amounts of protein loading were confirmed using β-actin. Data are expressed as the mean ± SD from a single triplicate experiment using ImageJ software. # *p* < 0.001 vs. untreated control group. * *p* < 0.05, ** *p* < 0.01, *** *p* < 0.001 vs. α-MSH-alone group.

**Figure 9 ijms-24-10393-f009:**
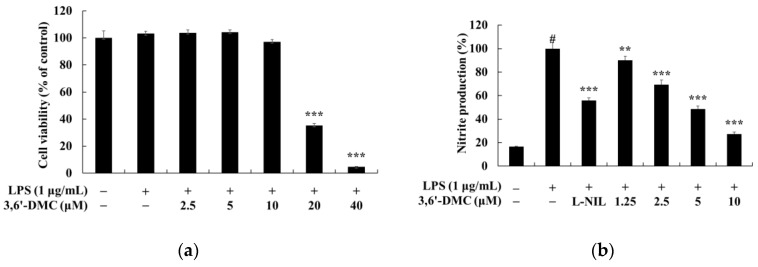
Effects of 3,6′-DMC on cell viability (**a**) and NO (**b**) production in LPS-induced RAW 264.7 cells. The cells were treated with 3,6′-DMC (2.5, 5, 10, 20, and 40 μM) in the presence of LPS (1 μg/mL) for 24 h. In these data, cell viability is expressed as the mean ± SD from three independent experiments, *** *p* < 0.001 vs. untreated control group. To check the effect of 3,6′-DMC on nitric oxide production in LPS-induced RAW 264.7 cells, the cells were treated with 3,6′-DMC (1.25, 2.5, 5, and 10 μM) in the presence of LPS (1 μg/mL) via stimulation for 24 h. L-NIL (40 μM) was used as a positive control. The inhibition of NO production by 3,6′-DMC treatment in LPS-induced RAW 264.7 cells was measured using the Griess reagent. In the data, NO oxide production is expressed as the mean ± SD from three repeated experiments. # *p* < 0.001 vs. untreated control group. ** *p* < 0.01, *** *p* < 0.001 vs. LPS-alone group.

**Figure 10 ijms-24-10393-f010:**
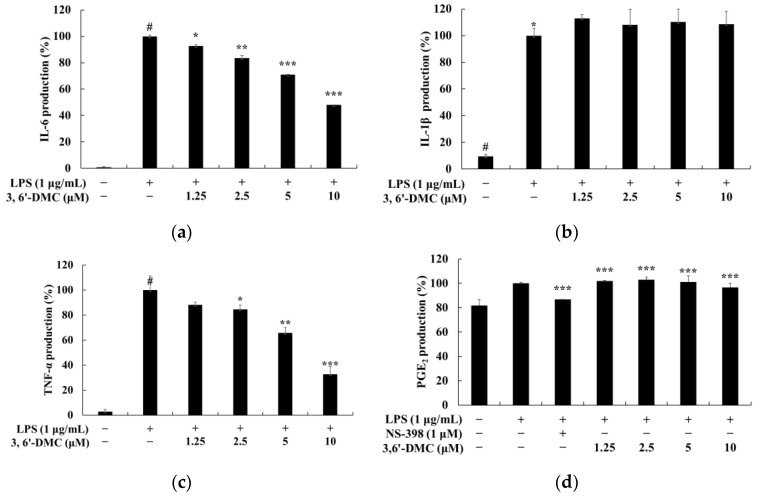
Effect of 3,6′-DMC on production of pro-inflammatory cytokines in LPS-induced RAW 264.7 cells. The cells were treated with 3,6′-DMC (1.25, 2.5, 5, and 10 μM) in the presence of LPS (1 μg/mL) via stimulation for 24 h. (**a**) IL-6 production, (**b**) IL-1β production, (**c**) TNF-α production, and (**d**) PGE_2_ production were determined using an ELISA kit. Data are expressed as the mean ± SD from a single triplicate experiment. # *p* < 0.001 vs. untreated control group. * *p* < 0.1, ** *p* < 0.01, *** *p* < 0.001 vs. LPS-alone group.

**Figure 11 ijms-24-10393-f011:**
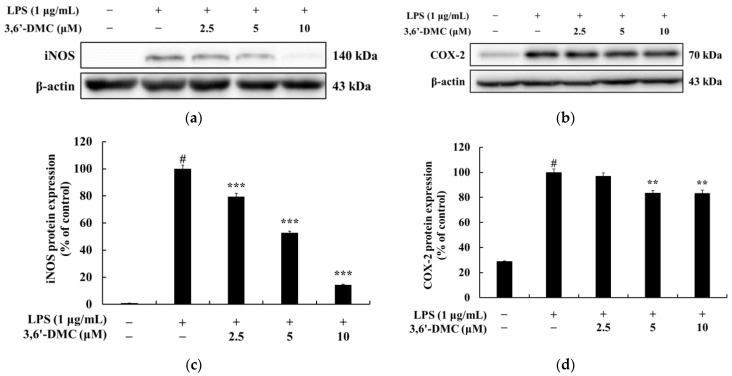
Effect of 3,6′-DMC on the protein expression of iNOS and COX-2 in LPS-induced RAW 264.7 cells. The cells were treated with 3,6′-DMC (2.5, 5, and 10 μM) in the presence of LPS (1 μg/mL) stimulation for 24 h. Western blotting results (**a**,**b**) and protein expression of iNOS/β-actin (**c**), COX-2/β-actin (**d**). Equal amounts of protein loading were confirmed using β-actin. Data are expressed as the mean ± SD from a single triplicate experiment using ImageJ software. # *p* < 0.001 vs. untreated control group. ** *p* < 0.01, *** *p* < 0.001 vs. LPS-alone group.

**Figure 12 ijms-24-10393-f012:**
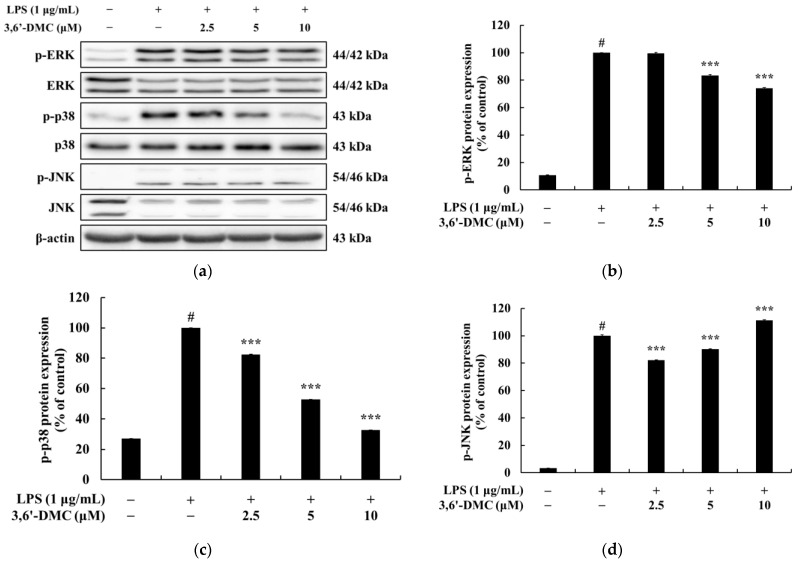
Effect of 3,6′-DMC on the phosphorylation expression of MAPK in LPS-induced RAW 264.7 cells. The cells were treated with 3,6′-DMC (2.5, 5, and 10 μM) in the presence of LPS (1 μg/mL) via stimulation for 20 min. (**a**) Western blotting results and protein expression of (**b**) p-ERK/ERK, (**c**) p-p38/p38, (**d**) p-JNK/JNK. Equal amounts of protein loading were confirmed using β-actin. Data are expressed as the mean ± SD from a single triplicate experiment using ImageJ software. # *p* < 0.001 vs. untreated control group. *** *p* < 0.001 vs. LPS-alone group.

**Figure 13 ijms-24-10393-f013:**
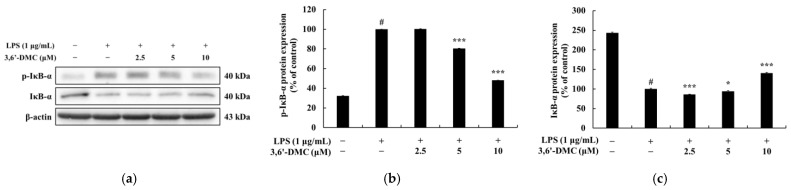
Effect of 3,6′-DMC on the protein expression of p-IκB-α and IκB-α in LPS-induced RAW 264.7 cells. The cells were treated with 3,6′-DMC (2.5, 5 and 10 μM) in the presence of LPS (1 μg/mL) via stimulation for 15 min. (**a**) Western blotting results and protein expression of (**b**) p-IκB-α/β-actin, (**c**) IκB-α/β-actin. Equal amounts of protein loading were confirmed using β-actin. Data are expressed as the mean ± SD from a single triplicate experiment using ImageJ software. # *p* < 0.001 vs. untreated control group. * *p* < 0.05, *** *p* < 0.001 vs. LPS-alone group.

**Figure 14 ijms-24-10393-f014:**
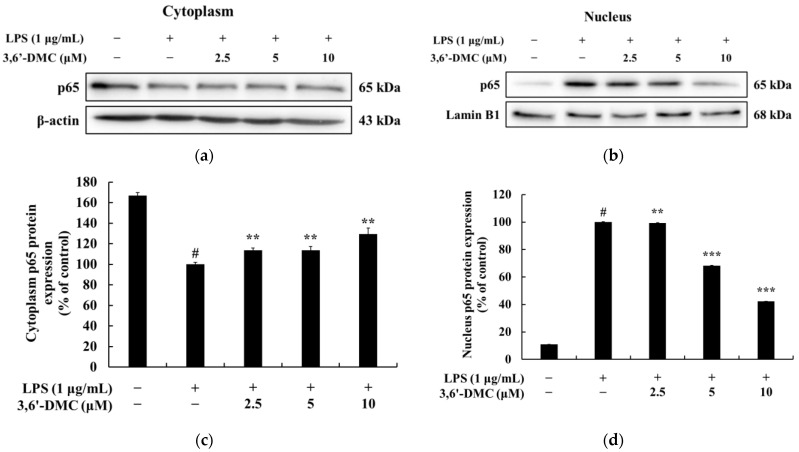
Effect of 3,6′-DMC on the protein expression of NF-κB in LPS-induced RAW 264.7 cells. The cells were treated with 3,6′-DMC (2.5, 5 and 10 μM) in the presence of LPS (1 μg/mL) via stimulation for 15 min. Protein expression of p65/β-actin in cytoplasm (**a**,**c**) and p65/Lamin B1 in nucleus (**b**,**d**). Equal amounts of protein loading were confirmed using β-actin and Lamin B1. Data are expressed as the mean ± SD from a single triplicate experiment using ImageJ software. # *p* < 0.001 vs. untreated control group. ** *p* < 0.01, *** *p* < 0.001 vs. LPS-alone group.

**Figure 15 ijms-24-10393-f015:**
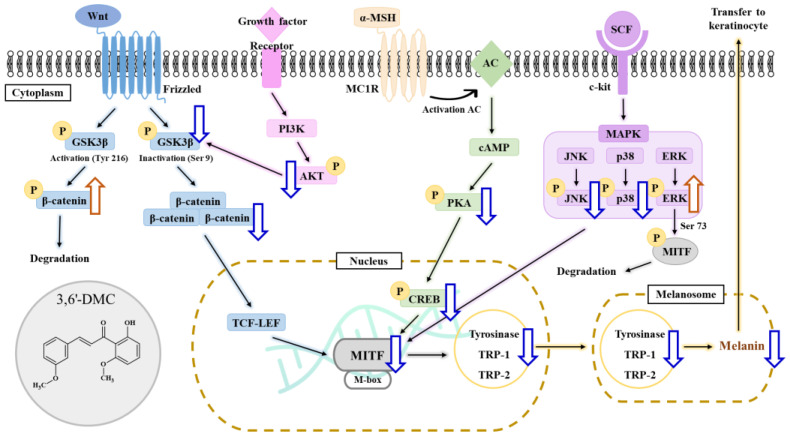
Schematic diagram of the proposed mechanism regulating the inhibitory action of 3,6′-DMC on melanogenesis.

**Figure 16 ijms-24-10393-f016:**
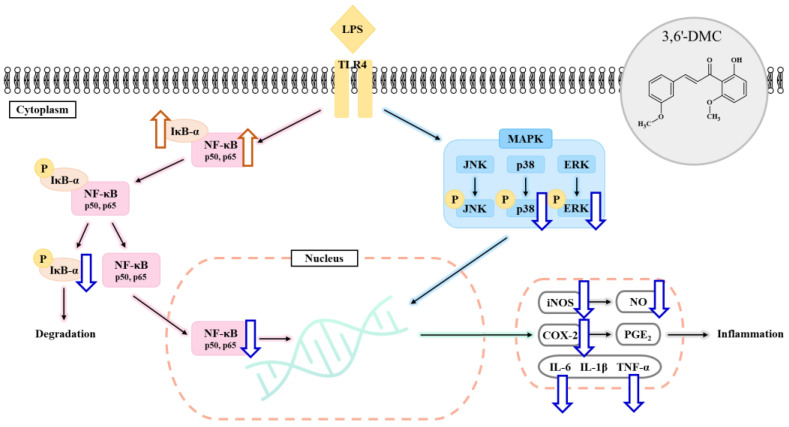
A schematic of the proposed mechanism of 3,6′-DMC in inhibiting LPS-stimulated inflammation in RAW264.7 cells. Acenocoumarol inhibits the MAPK and NF-κB pathway.

**Table 1 ijms-24-10393-t001:** Results of human skin primary irritation test (*n* = 33).

No	Test Samples	No. of Responder	24 h				48 h				Reaction Grade		
			+1	+2	+3	+4	+1	+2	+3	+4	24 h	48 h	Mean
1	3,6′-DMC (5 μM)	0	-	-	-	-	-	-	-	-	0	0	0
2	3,6′-DMC (10 μM)	0	-	-	-	-	-	-	-	-	0	0	0
3	Control (squalene)	0	-	-	-	-	-	-	-	-	0	0	0

## Data Availability

The data presented in this study are available on request from the corresponding author.

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
