# Peer review of "The Effects of 2-Hydroxy-3,6-Dimethoxychalcone on Melanogenesis and Inflammation"

_ijms, 2023, doi:10.3390/ijms241210393_

Round 1

Reviewer 1 Report

Comments on the article The Effects of 2′-Hydroxy-3,6′-Dimethoxychalcone on Melanogenesis and Inflammation” by Bae and Hyun.

The manuscript is interesting and written in a good English. The experimental design is also solid and well-constructed.

I consider that the manuscript can be published in this form after some minor revision. Please pay attention to the following suggestions:

I suggest clarifying the abbreviated words (Abstract section) if they are introduced for the first time in your manuscript, and you could arrange an abbreviations section before the Introduction section.

I suggest preparing the abstract in a quantitative manner; there is a lack of numerical results inside.

The quality of the figures is a little low, please improve them.

Author Response

<International Journal of Molecular Sciences>

< The Effects of 2′-Hydroxy-3,6′-Dimethoxychalcone on Melanogenesis and Inflammation>

Dear Editor,

Thank you for your useful comments and suggestions on the language and structure of our manuscript. We have modified the manuscript accordingly, and detailed corrections are listed below point by point. Our authors received English proofreading through native speakers before submitting paper (English-Editing-Certificate-67146, MDPI), and we will be done making corrections in English for the newly changed and inserted content paper (English-Editing-Certificate-XXXXX, MDPI):

Reviewer #1

  1. I suggest clarifying the abbreviated words (Abstract section) if they are introduced for the first time in your manuscript, and you could arrange an abbreviations section before the Introduction section.

→ We have inserted abbreviated words as you pointed out.

  1. I suggest preparing the abstract in a quantitative manner; there is a lack of numerical results inside.

→ We totally agree with your comments. On the other hand, we have been pointed out by another reviewer to reduce the word count of 309 in the abstract to around 200, which is the MDPI regulation, and it is difficult for us to insert additional information in a quantitative manner as we need to present our extensive research achievements on the melanin inhibitory and anti-inflammatory effects of 3,6'-DMC in the abstract. We ask for your generous understanding.

  1. The quality of the figures is a little low, please improve them.

→ We have increased the resolution of all pictures and corrected them as you pointed out.

Reviewer 2 Report

The authors have done extensive research on the e effects of 2′-hydroxy-3,6′-dimethoxychalcone (3,6’-DMC) on inflammation and melanogenesis. However, there are some questions that need to be addressed. These questions are somewhat critical and may affect the main conclusion of the paper. The authors should consider them seriously.

1.     Introduction: In this study, the Author(s) used 2′-hydroxy-3,6′-dimethoxychalcone (3,6'-DMC) purchased from chemical store, but in the second paragraph of the introduction specifically discusses Plant polyphenols which are less relevant to the purpose this research. If it must be loaded, also add the target compound from a synthetic perspective.

2.     Introduction: In paragraph 3, Author(s) only discussed that chalcone derived compounds are active for various bioactivities. Because in this study author (s) directly used the compound 2'-hydroxy-3,6'-dimethoxychalcone without synthesizing or isolating it first, Please add information on why the compound is active in terms of the framework or functional groups contained in the compound. So, the reason for choosing this compound is well illustrated.

3.     Discussion section: can the author predict the mechanism or pathway of how 3,6'-DMC inhibits melanogenesis and inflammation in the form of pictures or schematics. So that the data presented by the author(s) does not seem monotonous. 

Author Response

<International Journal of Molecular Sciences>

< The Effects of 2′-Hydroxy-3,6′-Dimethoxychalcone on Melanogenesis and Inflammation>

Dear Editor,

Thank you for your useful comments and suggestions on the language and structure of our manuscript. We have modified the manuscript accordingly, and detailed corrections are listed below point by point. Our authors received English proofreading through native speakers before submitting paper (English-Editing-Certificate-67146, MDPI), and we will be done making corrections in English for the newly changed and inserted content paper (English-Editing-Certificate-XXXXX, MDPI):

Reviewer #2

  1. Introduction: In this study, the Author(s) used 2′-hydroxy-3,6′-dimethoxychalcone (3,6'-DMC) purchased from chemical store, but in the second paragraph of the introduction specifically discusses Plant polyphenols which are less relevant to the purpose this research. If it must be loaded, also add the target compound from a synthetic perspective.

→ In the introduction, we tried to present polyphenols and the importance of the chalcones in these compounds, and during the revision, at the request of other reviewers, we reinforced the superiority and necessity of natural products over synthetics.

  1. Introduction: In paragraph 3, Author(s) only discussed that chalcone derived compounds are active for various bioactivities. Because in this study author (s) directly used the compound 2'-hydroxy-3,6'-dimethoxychalcone without synthesizing or isolating it first, Please add information on why the compound is active in terms of the framework or functional groups contained in the compound. So, the reason for choosing this compound is well illustrated.

→In the introduction, we tried to present polyphenols and the importance of the chalcones in these compounds, and during the revision, at the request of other reviewers, we reinforced the superiority and necessity of natural products over synthetics.

  1. Discussion section: can the author predict the mechanism or pathway of how 3,6'-DMC inhibits melanogenesis and inflammation in the form of pictures or schematics. So that the data presented by the author(s) does not seem monotonous.

→We have redrawn Figure 15 and Figure 16 for the mechanism pathway by which 6'-DMC inhibits melanogenesis and inflammation as per your recommendation.

Reviewer 3 Report

The current manuscript is an interesting study on the effects of a novel chalcone derivative on melanogenesis and inflammation, comprising several studies and including a mechanistic approach. It is overall well done and complete, nevertheless, some alterations should be made before acceptance for publication:

- The abstract is too long for MDPI rules: it can have a maximum of 200 words, hence authors should summarize it;

- In the introduction section, more should be said about the limitations of current treatments for melanoma and inflammatory skin diseases, in order to better support the concept that new compounds are necessary; here, authors should further describe the skin, anatomically, physiologically and chemically, and address the difficulties in topical or transdermal drug delivery;

- Additionally, also in the introduction section more should be said about the general advantages of plant derived compounds when compared to synthetic entities;

- Authors should also mention specific examples of “citrus fruits, vegetables, and spices” in which chalcones are present;

- Figure quality should be improved, consider restructuring the graphs to make them individually bigger, so that words are more easily read;

- Abbreviations should always be defined in the text before being used, authors should check and correct this (for example “MTT”);

- A conclusion section, separate from the discussion section, should exist;

- For statistical analysis, the used software should be mentioned;

- How are the authors aiming to formulate the studied compound in the future? A cream, a gel, a suspension, a solution? What problems might arise, related to the compounds solubility, vulnerability to chemical or metabolic degradation etc.?; are the authors considering nanosystems as an option? And if yes, which types and compositions? This should be addressed in the manuscript.

Author Response

<International Journal of Molecular Sciences>

< The Effects of 2′-Hydroxy-3,6′-Dimethoxychalcone on Melanogenesis and Inflammation>

Dear Editor,

Thank you for your useful comments and suggestions on the language and structure of our manuscript. We have modified the manuscript accordingly, and detailed corrections are listed below point by point. Our authors received English proofreading through native speakers before submitting paper (English-Editing-Certificate-67146, MDPI), and we will be done making corrections in English for the newly changed and inserted content paper (English-Editing-Certificate-XXXXX, MDPI):

Reviewer #3

  1. The abstract is too long for MDPI rules: it can have a maximum of 200 words, hence authors should summarize it;

→ We have reduced the word count of the abstract by nearly 100 characters as you pointed out.

  1. In the introduction section, more should be said about the limitations of current treatments for melanoma and inflammatory skin diseases, in order to better support the concept that new compounds are necessary; here, authors should further describe the skin, anatomically, physiologically and chemically, and address the difficulties in topical or transdermal drug delivery;

→I've included your point in the conclusion as future research

  1. Additionally, also in the introduction section more should be said about the general advantages of plant derived compounds when compared to synthetic entities;

→ As you pointed out, I've added a section in the introduction about the benefits of "natural products"(Line 59-71)

  1. Authors should also mention specific examples of “citrus fruits, vegetables, and spices” in which chalcones are present;.

→ As you pointed out, we made the following modifications. : “such as fruits (e.g., citruses, apples), vegetables (e.g., tomatoes, shallots, bean sprouts, potatoes) and various plants and spices (e.g., licorice)”

  1. Figure quality should be improved, consider restructuring the graphs to make them individually bigger, so that words are more easily read;.

→ We have increased the resolution of all pictures and corrected them as you pointed out

  1. Abbreviations should always be defined in the text before being used, authors should check and correct this (for example “MTT”

→ We have inserted abbreviated words as you pointed out

  1. A conclusion section, separate from the discussion section, should exist;

→ We have inserted a "conclusion section" as you pointed out.

  1. For statistical analysis, the used software should be mentioned.;

→ We have presented the software used for statistical analysis as you pointed out

  1. How are the authors aiming to formulate the studied compound in the future? A cream, a gel, a suspension, a solution? What problems might arise, related to the compounds solubility, vulnerability to chemical or metabolic degradation etc.?; are the authors considering nanosystems as an option? And if yes, which types and compositions? This should be addressed in the manuscript.

→ In the "conclusion section" we inserted the following sentence. : Although the stability of 3,6'-DMC has been confirmed, formulation studies with nano vehicles such as exosomes are needed to address issues such as the solubility of 3,6'-DMC and its vulnerability to chemical or metabolic degradation. 
